# Molecular Iodine/Cyclophosphamide Synergism on Chemoresistant Neuroblastoma Models

**DOI:** 10.3390/ijms22168936

**Published:** 2021-08-19

**Authors:** Winniberg Álvarez-León, Irasema Mendieta, Evangelina Delgado-González, Brenda Anguiano, Carmen Aceves

**Affiliations:** Instituto de Neurobiología, Universidad Nacional Autónoma de México (UNAM) Juriquilla, Querétaro 76230, Mexico; winnsteph@gmail.com (W.Á.-L.); aliciairasema@hotmail.com (I.M.); edelgado@comunidad.unam.mx (E.D.-G.); anguianoo@unam.mx (B.A.)

**Keywords:** neuroblastoma, molecular iodine, cyclophosphamide, xenografts, metronomic therapy

## Abstract

Neuroblastoma (Nb), the most common extracranial tumor in children, exhibited remarkable phenotypic diversity and heterogeneous clinical behavior. Tumors with MYCN overexpression have a worse prognosis. MYCN promotes tumor progression by inducing cell proliferation, de-differentiation, and dysregulated mitochondrial metabolism. Cyclophosphamide (CFF) at minimum effective oral doses (metronomic therapy) exerts beneficial actions on chemoresistant cancers. Molecular iodine (I_2_) in coadministration with all-trans retinoic acid synergizes apoptosis and cell differentiation in Nb cells. This work analyzes the impact of I_2_ and CFF on the viability (culture) and tumor progression (xenografts) of Nb chemoresistant SK-N-BE(2) cells. Results showed that both molecules induce dose-response antiproliferative effects, and I_2_ increases the sensibility of Nb cells to CFF, triggering PPARγ expression and acting as a mitocan in mitochondrial metabolism. In vivo oral I_2_/metronomic CFF treatments showed significant inhibition in xenograft growth, decreasing proliferation (Survivin) and activating apoptosis signaling (P53, Bax/Bcl-2). In addition, I_2_ decreased the expression of master markers of malignancy (MYCN, TrkB), vasculature remodeling, and increased differentiation signaling (PPARγ and TrkA). Furthermore, I_2_ supplementation prevented loss of body weight and hemorrhagic cystitis secondary to CFF in nude mice. These results allow us to propose the I_2_ supplement in metronomic CFF treatments to increase the effectiveness of chemotherapy and reduce side effects.

## 1. Introduction

Neuroblastoma (Nb) is the most common extracranial tumor in children accounting for 15% of pediatric oncology deaths. The overexpression of the neural MYC gene (MYCN) characterizes the chemoresistant Nb and has a worse prognosis [1]. MYCN induces cell proliferation, inhibits cell differentiation, and maintains the stem-like phenotype [2]. These levels correlate to metastasis and angiogenesis, and MYCN overexpression affects the mitochondria metabolism to support the higher energy demand of chemoresistant Nb cells [1,2]. One of the most widely used drugs in this pathology is cyclophosphamide (CFF), an effective and low-cost chemotherapy. CFF is a prodrug, biotransformed into two metabolites: phosphoramide mustard (amino-[bis(2-chloroethyl)amino] phosphinic acid), which is the active antineoplastic principle, and acrolein (prop-2-enal), associated with several side effects including inflammation and hemorrhagic cystitis [3]. The use of CFF in metronomic therapy has recently been proposed as an effective alternative for chemoresistant cancers [4]. Metronomic therapy uses chronic oral treatments with minimum effective doses that exert their effects by inhibiting angiogenesis, immune modulation of the tumor stroma, and apoptosis of tumor cells. Its long-term and low-dose use reduces side effects and maintains the patient’s quality of life [4]. Furthermore, the use of combined therapies that include cellular re-differentiation or immune reactivation messengers are promising approaches that are currently being tested [5,6].

In addition, the antineoplastic effects of molecular iodine (I_2_) are well established, triggering apoptotic and redifferentiation mechanisms in several cancer cells, including mammary, ovary, and prostate, among others [7]. These effects are mediated partially by the activation of peroxisome proliferator-activated receptors type gamma (PPARγ) and directly, as a mitocan element, by thiol depletion and disruption of the mitochondrial membrane potential (Mmp), triggering the intrinsic apoptosis [8]. Furthermore, our previous report showed that the I_2_ supplement sensitized Nb cells to all-trans retinoic acid (ATRA) in vitro and synergized the antitumor effect of ATRA preventing body-weight loss and diarrhea episodes in nude mice [9]. The present work analyzes the impact of I_2_ and metronomic CFF supplements on the viability (culture) and tumor progression (xenografts) of chemoresistant neuroblastoma SK-N-BE(2) cells [10].

## 2. Results

### 2.1. Results In Vitro

#### 2.1.1. Viability

Figure 1A shows the effect of molecular iodine (I_2_), CFF, and their combination on the viability of the SH-SY5Y and SK-N-BE(2) cell lines at 96 h. Both components generate similar dose-response action, being SH-SY5Y the most sensitive and corroborating that SK-N-BE(2) is a highly resistant cell line. I_2_ supplementation showed an IC50 of 205.5 µM in SH-SY5Y cells and 343.7 µM in SK-N-BE(2) cells (Appendix A). In CFF, the IC50 was 0.602 µM in SH-SY5Y and 1.045 µM in SK-N-BE(2) (Appendix A). Specific time-response data are summarized in Appendix A. All further experiments were analyzed in SK-N-BE(2) cells. The combination of 200 µM I_2_ with two concentrations of CFF confirmed that the presence of iodine increases the sensibility of these cells to CFF, improving response by 15% at 1.0 µM CFF and up to 40% at 0.5 µM CFF (Figure 1B). The combination index (CI) for I_2_/CFF treatments presented a CI value of 0.00835 at 0.5 µM CFF and 3.39 at 1 µM CFF, indicating synergism at 0.5 µM CFF. To analyze the participation of PPARγ in the I_2_ response, we used the agonist rosiglitazone (RGZ 5.0 µM) in the presence or absence of the antagonist GW9662 (1.0 µM) for 96 h. RGZ showed a similar inhibitory effect in viability observed with I_2_ (34 vs. 28%). The preincubation (2 h before) with GW9662 canceled the RGZ inhibitor effect but had a partial impact on the I_2_ supplement (24%), suggesting that I_2_ exerts its actions through other mechanisms besides PPARγ (Figure 1C).

#### 2.1.2. Apoptosis

Figure 2 shows the percentage of apoptosis-positive cells (annexin Cy5) at two different times. At 48 h, 200 µM I_2_ and 1.0 µM CFF exhibited similar apoptotic induction (~35% each), and the adjuvant action (I_2_ + CFF group) enhanced the apoptosis effect to 40%. At 96 h, apoptosis was maintained primarily in I_2_ groups. The apoptotic index Bax/Bcl-2 (RT-PCR), an indicator of caspase pathway activation, indicated that both components exert significant induction, showing a predominant action of I_2_ at both times.

#### 2.1.3. Mitochondrial Activity

Figure 3 depicts the fluorescence generated by the MitoTracker entrance after I_2_ and CFF treatments. Iodine exerted a rapid and significant increment in mitochondrial permeability at 12 h and maintained until 48 h. In contrast, CFF induced a change in mitochondria metabolism after 48 h. The presence of both components exhibited a synergistic effect only at 48 h.

#### 2.1.4. Molecular Response

Figure 4 shows the effect of treatments on gene expression. I_2_ supplements increased the genes associated with differentiation (PPARγ) and diminished those associated with aggressiveness (MYCN) and resistance (Survivin; SVV). The presence of CFF only reduced SVV expression. In the I_2_ + CFF group, only the effects of I_2_ remained, suggesting that I_2_ is the inductor of differentiation. The multidrug resistance gene MDR1 did not exhibit differences with any treatment.

### 2.2. Results In Vivo

#### 2.2.1. Tumor Growth

Nude mice with SK-N-BE(2) xenografts were used to analyze the effects of I_2_ and CFF under in vivo conditions. Treatments were given when the tumor reached a size of 1.5 cm^3^. We evaluated the impact of oral metronomic CFF doses (20 mg/kg/day; 0.06%) with or without the I_2_ supplement (8 mg/kg/day; 0.025%) for three weeks. Previous studies from our laboratory found that the presence of xenografts generates a small but consistent reduction in body weight gain (BWG) in all animals and that the I_2_ supplement prevents this loss. Figure 5 shows that I_2_-supplemented animals exhibited an increase in BWG like that of the control group of xenograft-free (x.f.) animals. The animals with tumors, both the control and the CFF groups, showed a reduction in BWG from the first week. In contrast, the combined group (I_2_ + CFF) recovered BWG in the last week of treatment, indicating a beneficial effect of I_2_.

Concerning tumor progression, after the third week, all treatments significantly inhibited tumor growth. The tumor volume decreased 44.72% with the I_2_ supplement and 68.11% with CFF. The coadministration of both components (I_2_ + CFF) showed a considerable, yet not statistically significant, tumor inhibition, decreasing the tumor size by 78.78%.

A characteristic of these types of tumors is their aberrant vascularization and abundant bleeding, and so they are known as blue tumors. As shown in Figure 5, I_2_ supplementation considerably decreased bleeding patterns regardless of tumor size. However, the CFF group alone showed a decrease in tumor size but did not change bleeding appearance.

#### 2.2.2. Histopathology

Figure 6 shows the analysis of vascularization (H&E stain and CD34 immunohistology and quantification) and collagen fibrosis content (Masson’s stain and quantification). Control and CFF xenograft micrography showed aberrant vascular patterns and abundant extravascular erythrocytes (H&E stain). In contrast, treatments with I_2_ alone and combined (I_2_ + CFF) showed a consistent reduction in the vasculature, minor invasion of tumor cells in vascular structures, and fewer extravascular erythrocytes. The quantification of mean vascular density (MVD) by CD34 immunohistochemistry showed that I_2_ and CFF decrease vasculature area in comparison with control group. A significant increase in positive collagen fibers (blue stain) is observed in I_2_ and CFF groups suggesting a hypoxic microenvironment, with tumoral cell death fibrosis substitution. Following this hypoxic status, I_2_-treated tumors exhibited an elevated expression of Hypoxia-inducible factor (HIF1) but not vascular endothelial growth factor (VEGF), which corroborates that the I_2_ treatments do not induce an increase in vascularization.

#### 2.2.3. Molecular Response

Similar to the in vitro results, xenografts from the control animal expressed an elevated amount of MYCN (Figure 7). The I_2_ supplement modified the xenografts’ aggressivity pattern, increasing the expression of differentiation promoters (PPARγ and TrkA) and decreasing those related to resistance (MYCN and TrkB). Interestingly, the higher TrkB expression promoted by the CFF supplement was suppressed by the presence of I_2_ (I_2_ + CFF group). In addition, both components (I_2_ and CFF) induced the expression of p53 and increased the apoptotic index Bax/Bcl-2.

#### 2.2.4. Preventive Effect of I_2_ in Bladder Damage

The more frequent side effect of CFF treatment is hemorrhagic cystitis, evidenced by hypervascularity, edema, inflammation, and bleeding. We examined the bladder morphology at the end of the experiment and evaluated the vasculature (blood vases/field) through expression of CD34 (immunohistochemistry) and histopathology (H&E) (Figure 8). No significant differences in the vasculature were found in the treatments. However, the stains with H&E revealed clear signs of edema in the lamina propria and an increased thickness in the urothelium in the CFF group (arrows). I_2_ and I_2_ + CFF group bladders did not show these alterations, indicating the I_2_ alone did not cause any irritation and that, in the presence of CFF, this halogen exerts a significant preventive action.

## 3. Discussion

Previously our group showed that I_2_ in coadministration with ATRA synergizes apoptosis and cell differentiation in Nb cells [9]. The present work analyzes the impact of I_2_ and CFF in viability and tumor progression of Nb chemoresistant SK-N-BE(2) cells. Our results corroborate that these cells have a high resistance to various antineoplastic components since both I_2_ and CFF exert attenuated effects up to 40% compared to the more sensitive SH-SY5Y cells [10]. However, an important finding is that the I_2_ supplement, which does not generate any side effects, increased the sensitivity to CFF by 25 to 40%. It is well established that the primary mechanism of CFF is the induction of p53 apoptosis via DNA adducts formation [11]. Conversely, I_2_ actions are more complex since this halogen could act directly on mitochondria by inducing an apoptosis cascade [12], or indirectly by activating PPARγ and triggering redifferentiation or apoptotic signaling [7,13]. Our results showed that both components (I_2_ and CFF) increased apoptosis (exposure of annexin-Cy5 and low expression of SVV), but only I_2_ groups modified master differentiation genes, decreasing the expression of MYCN, and significantly inducing PPARγ. Iodine in neoplastic cells binds with arachidonic acid and generates an iodolipid called 6-iodolactone (6-IL) [14,15]. This iodolipid is a specific activator of PPARγ [16,17]. These receptors are expressed in Nb, and the use of agonists impairs proliferation and induces differentiation of these cells [9,18]. PPARγ agonists reduce levels of MYCN by inhibiting critical molecules within the PI3K/AKT/mTOR signaling pathway increasing GSK-3β activity as well as MYCN phosphorylation and its proteasome degradation [19,20,21]. We corroborated that PPARγ reduces the viability of these SK-N-BE(2) Nb cells since the supplement with 1 uM RGZ decreased its proliferation. Moreover, we found that the I_2_ effect was partially canceled with the agonist GW9662, suggesting that the antineoplastic effects of I_2_ include the activation of PPARγ, but other mechanisms also contribute.

Mitochondria metabolism is considered a hallmark of cancer, showing a crucial contributor in the process as metabolic reprogramming, generation of reactive oxygen species (ROS) and production of metabolites that enhance oncogenesis [22]. Recently, several therapeutic approaches for these processes identified the “mitocans,” a category of drug that targets the mitochondria of cancer cells [8]. Many natural agents can target mitochondria and exert anticancer activities with minimal or no side effects. In the light of this process, I_2_ seems to be a mitocan since, in cancer cells, I_2_ depleted thiol generation and disrupted the Mmp, inducing significant increases in its permeability and triggering apoptosis [12,23]. The substantial rise in MitoTracker signaling observed in I_2_ groups starting at 12 h corroborated these direct mitochondrial effects and explains, in part, the increase in CFF sensibility. Previous data from our laboratory showed that this increase in Mmp was accompanied by a decrease in SVV content in both intramitochondrial and cytoplasmic compartments [9]. SVV is an apoptosis-inhibiting factor (IAP) [24]. SVV overexpression in Nb cells makes them resistant to ATRA, protecting them against agents that damage DNA, and stabilizes the mitochondrial membrane by decreasing apoptosis induction [20,25].

The xenograft model was efficient and reproducible since we obtained between 95 and 100% implantation. Moreover, we corroborated that this is an aggressive Nb type generating fast-growing tumors and hypervascularization characteristics (bluish). The results showed that both the metronomic CFF and the I_2_ supplement effectively decreased tumor growth (68.11% and 44.72%, respectively), especially when both components were co-administered (78.78%). In addition, the I_2_ supplement was accompanied by decreased aberrant vascularization and bleeding, associated with a significant increase in the expression of HIF1, which is the first signaling secondary to lack of oxygen. No change in the expression of VEGF, the inducer of new vessels, was observed. This pattern can be interpreted as a modification in the vascular cytoarchitecture rather than a process of angiogenesis, but the possible mechanisms involved in this I_2_ effect have not yet been elucidated. However, conventional mechanisms of I_2_ could participate. Angiogenesis is an active process that involves a significant increase in ROS generation, promoting the angiogenic switch from quiescent to active endothelial cells [25,26]. It is possible that the mitocan effect of iodine neutralized ROS [8]. An alternative way might be the significant decrease in MYCN expression observed with I_2_ treatments. MYCN induces angiogenesis by its direct action in VEGF amplification [2].

At molecular level, the decrease in tumor size was accompanied by an increase in the apoptosis markers p53 and Bax/Bcl-2 index in all treated groups. This apoptotic induction also agrees with the finding that the tumors supplemented with both components had a higher proportion of type 1A collagen fibers (Masson’s trichrome staining), indicating the replacement of epithelial cancer cells with fibrous tissue in response to rapid induction of apoptosis.

In addition, one important finding is the differential gene response between tumors treated with I_2_ vs. CFF. Our results agreed with the in vitro results, that only I_2_ supplements exert an evident modulation in the differentiation master genes increasing PPARγ and TrkA, decreasing the basal expression of MYCN and TrkB, and canceling the increase of TrkB secondary to CFF treatment. MYCN amplification in Nb is typically associated with epigenetic abnormalities to impair apoptosis, followed by the overexpression of the anti-apoptotic proteins Bcl-2, SVV, and TrkB [1,19,20]. TrkB stimulates cell survival and angiogenesis and activates the survival pathway PI3K/AKT, contributing to increased drug resistance [26,27]. Therefore, the rise in TrkB in the presence of CFF could be interpreted as a response of tumor cells to the drug’s toxic effect. Studies analyzing this hypothesis are needed; however, the prevalence of I_2_ action indicates an antitumor benefit.

Finally, we explored the possible role of the I_2_ supplement in the side effects prevention associated with xenograft signalization and the bladder injury secondary to CFF administration. Previous studies in our group had detected that xenografts and tumor growth generate stress in the mouse, evidenced in the loss of BWG [9]. This effect has been described in preclinical and clinical studies and is known as cachexia [28]. Cachexia is accompanied by loss of adipocytes and muscle tissue with chronic inflammation and increases in proinflammatory factors such as TNFα and IL-6 [28,29]. The prevention of weight loss in the I_2_-supplemented groups might be due to two conditions: first, the antineoplastic effect of I_2_ that prevents tumor growth and decreases the tumor mass signaling; and second, a direct impact on chronic inflammation processes due to I_2_ antioxidant action [9,30]. This effect also appears to be exercised in the prevention of bladder injury. It is well known that the hepatic biotransformation of CFF produces, in addition to phosphoramide mustard (an antineoplastic metabolite), acrolein, which causes hemorrhagic cystitis [3]. We did not expect severe effects of acrolein at the metronomic dose used; however, the histological evaluation showed a thickening of the urothelium and the lamina propria of the bladder, which indicates moderate hemorrhagic cystitis. The I_2_ supplement in coadministration with CFF prevented these alterations. This protective mechanism might be due to its antioxidant action. Previously, it has been shown that the chemical form of I_2_ has an in vitro reducing capacity (FRAP test) ten times greater than ascorbic acid and 60 times greater than potassium iodide [31]. In vivo studies showed that the iodine supplement decreases the oxidative potential in the serum of rodents and patients [30]. The administration of other antioxidants, such as ascorbic acid, retinol, and resveratrol, improves oxidative stress by reducing ROS levels in bladder tissues generated by CFF treatment [32,33]. An unexplored alternative is that the I_2_ could be binding directly to acrolein, decreasing its irritating action, and preventing its contact or entry into the urothelium of the bladder. This alternative is based on the acrolein structure that contains double bonds capable of being iodinated [34].

## 4. Materials and Methods

### 4.1. Chemicals and Reagents

The Cyclophosphamide for in vivo assays and the CFF active metabolite 4-Hydroperoxycyclophosphamide for in vitro assays were obtained by Cryofarma (Jalisco, Mexico) and Toronto Research Chemicals (Toronto, Ontario, CA, USA) respectively, and we denominate both with the same abbreviation (CFF). Rosiglitazone (RGZ; PPARγ-specific agonist, by Cayman Chemical, Los Angeles, CA, USA), GW9662 (PPARγ-specific antagonist, by Corning, Bedford, MA, USA) and Matrigel (basement membrane matrix, Corning, Bedford, MA, USA). Sublimed iodine was obtained from Macron-Avantor (Center Valley, PA, USA). The concentration of iodine solutions was verified by sodium thiosulfate titration. All other chemicals were of the highest purity grade available.

### 4.2. Cell Culture

The Nb cell lines SH-SY5Y (CRL-2266) and SK-N-BE(2) (CRL-2271) were obtained from the company American Type Culture Collection (ATCC, Manassas, VA, USA). All the experiments were performed with passages 1–5 and recently tested and authenticated by STR profiling (BIMODI Invoice number 190320-029). The conditions for cell culture were Dulbecco’s Modified Eagle’s Medium (DMEM) supplemented with fetal bovine serum (FBS, 10%) and penicillin/streptomycin (2%) by Invitrogen (Carlsbad, CA, USA) in a humidified chamber with 5% CO_2_ atmosphere and 95% air at 37 °C.

### 4.3. Cell Viability

A total of 50,000 cells/well were seeded onto 12-well plates. After 24 h, different concentrations of CFF (0.5, 1, and 2 µM), I_2_ (100, 200 and 400 µM) and I_2_ + CFF (200 + 0.5 or 200 + 1.0, respectively) were added for 0, 24, 48, 72, and 96 h. Control groups were followed at the same times using deionized water as treatment (vehicle of I_2_ and CFF). In the GW9662/RGZ or I_2_-treated groups, GW9662 (1 µM) was administered 2 h before RGZ (5 µM) or I_2_ (200 µM) treatment.

After treatment, cells were detached and mixed with the exclusion dye trypan blue (0.04%) to count the cells using a hemocytometer in light microscopy; viability was reported as fold change against control. All experiments were carried out in three independent experiments per triplicate. To measure the extent of interaction between I_2_ and CFF, data were analyzed by CompuSyn software 1.0 (ComboSyn, Inc., Paramus, NJ, USA) based on the combination index (CI) of the multiple drug effect equation of Chou-Talalay [35].

### 4.4. Apoptosis

Apoptosis was evaluated by flow cytometry using the apoptosis kit (ABCAM No. 14190, Cambridge, UK) with the Attune NXT flow cytometer (BRVY). Briefly, the pellet of cells was resuspended in PBS, and the monoclonal antibody for annexin and propidium iodide were added, according to manufacturer’s instructions. The mixture was incubated for 30 min at room temperature and protected from light. After incubation, the cells were washed twice with PBS and resuspended in 500 µL PBS. Data analysis of 10,000 events was performed using FlowJo v10 (Trial version) software.

### 4.5. Gene Expression

TrkA, TrkB, PPARγ, SVV, MDR-1, MYCN, P53, Bax, Bcl-2, VEGF, HIF1, and β-actin were analyzed by RT-qPCR from SK-N-BE(2) cell cultures and xenografts after the corresponding treatments. Briefly, total RNA was obtained using Trizol reagent (Life Technologies, Inc., Carlsbad, CA, USA). RNA (2 µg) was reverse transcribed (RT) using oligo-deoxythymidine (Invitrogen, Waltham, MA, USA). Real-time PCR was performed on the Rotor-Gene 3000 sequence detector system (Corbett Research, Mortlake, NSW, Australia) using SYBR Green as a DNA amplification marker (gene-specific primers are listed in Table 1). Relative mRNA levels were normalized to the mRNA level of β-actin.

### 4.6. Mitochondrial Membrane Potential

After 12, 24, and 48-h treatments (I_2_, CFF, or I_2_ + CFF), the cells were PBS-washed and labeled with 200 nM MitoTracker Red CM-H2Xros (Thermo Fisher; Waltham, MA, USA) for 45 min. Then, cells were fixed for 10 min with ethanol, PBS-washed, and mounted with anti-FADE and DAPI. Micrographs were taken with an epifluorescence microscope (Axio Imager, Carl Zeiss, Jena, Germany). The software Image J 1.8 (National Institutes of Health, Bethesda, MD, USA) was used to quantify the relative fluorescence units (RFUs) and determine the mitochondrial functional state.

### 4.7. Tumoral Implantation and Progression

Xenografts with SK-N-BE(2) cells were generated using 5 × 10^6^ cells/injection of Nb cells in 6–7-week-old male immunodeficient athymic nude mice (Foxn1 nu/nu, Harlan Mexico, Ciudad de Mexico, Mexico) as previously described [36]. Mice were housed in barrier conditions under a 12-h light/dark cycle with food and water supplied ad libitum. All the procedures followed the Animal Care and Use Program of the National Institutes of Health (NIH; Bethesda, MD, USA) and were approved by the Ethics Committee of the Instituto de Neurobiología (ethical approval number 035).

When palpable tumors reached a volume of 1 cm^3^, animals were randomly assigned to each group (*n* = 4). I_2_ (8 mg/kg/day; 0.025%), CFF (20 mg/Kg/day; 0.060%), or a mixture of both. The treatments were supplied in drinking water ad libitum. The control group received only water. Animals were sacrificed after anesthesia with a ketamine/xylazine mixture (30 mg/Kg and 6 mg/Kg from Pisa Agropecuaria, Hgo., Mexico, and Cheminova CDMX, Mexico, respectively). The bladder and a tumor section were fixed in 10% formalin for at least 24 h and processed for immunohistochemistry. The remaining tumors were frozen in dry ice for RNA analysis.

### 4.8. Immunohistochemistry

Tumor sections and bladders were stained with hematoxylin-eosin (H&E) and Masson’s trichrome techniques for histopathological analysis. In addition to the vasculature analysis, the endothelial protein antibody CD34 (ab182981; 1:2500, Abcam, Cambridge, UK) was used to detect endothelial-positive cells (Vector Labs, Burlingame, CA, USA). Sections were counterstained by hematoxylin. The Mean Vascular Density (CD34/mm^2^) or vascular number per field were quantified by randomly analyzing three fields from three different sections of each tumor and bladder, using the software ImageJ version 1.8 (National Institutes of Health, Bethesda, MD, USA).

### 4.9. Western Blot

Western blot (WB) analysis of PPARγ proteins for tumor tissue was performed with the chemiluminescence technique [9]. Briefly, 50 µg of protein per lane were separated by electrophoresis in 10% acrylamide gel, proteins were later transferred to a nitrocellulose membrane (Bio-Rad, Hercules, CA, USA). The unspecified reaction was blocked overnight with PBS containing 5% skimmed milk powder. The membranes were treated with polyclonal antibodies (Santa Cruz Biotechnology, Los Angeles, CA, USA) against anti-PPARγ (ab209350, 54 kDa, 1: 1000, Abcam, Cambridge, MA, USA). As a secondary antibody, goat anti-rabbit (Thermo scientific 656120, 1: 10,000, Invitrogen, Waltham, MA, USA) was used. Proteins were visualized using chemiluminescent detection (ECL, Amersham Biosciences, Buckinghamshire, UK). The blots were visualized and pictured with Image LabTM (Bio-rad), and the densitometry analysis was performed with Image ImageJ V1.53e; PPAR levels were normalized to total protein of Ponceau red staining signal [37].

### 4.10. Statistical Analysis

Data for in vitro experiments are the media of three independent tests in triplicate. In vivo, four animals per group were used. Tissue analysis for PCR is the average of four samples, and three sections of each tumor were used for immunohistochemistry. Statistical analysis was performed by one-way ANOVA followed by Tukey’s test for analysis between groups. Values with *p* < 0.05 were considered statistically significant.

## 5. Conclusions

Molecular iodine exerts antiproliferative and differentiation effects in Nb cell lines, increasing their sensitivity to CFF. Molecular mechanisms include decreased expression of master regulators related to malignancy (MYCN, TrkB), remodeling of the vasculature, and increased differentiation signaling (PPARγ and TrkA). Furthermore, I_2_ supplementation prevents loss of body weight and hemorrhagic cystitis secondary to CFF in nude mice. These results allow us to propose the I_2_ supplement in metronomic CFF treatments to increase the effectiveness of chemotherapy and reduce side effects.

## Figures and Tables

**Figure 1 ijms-22-08936-f001:**
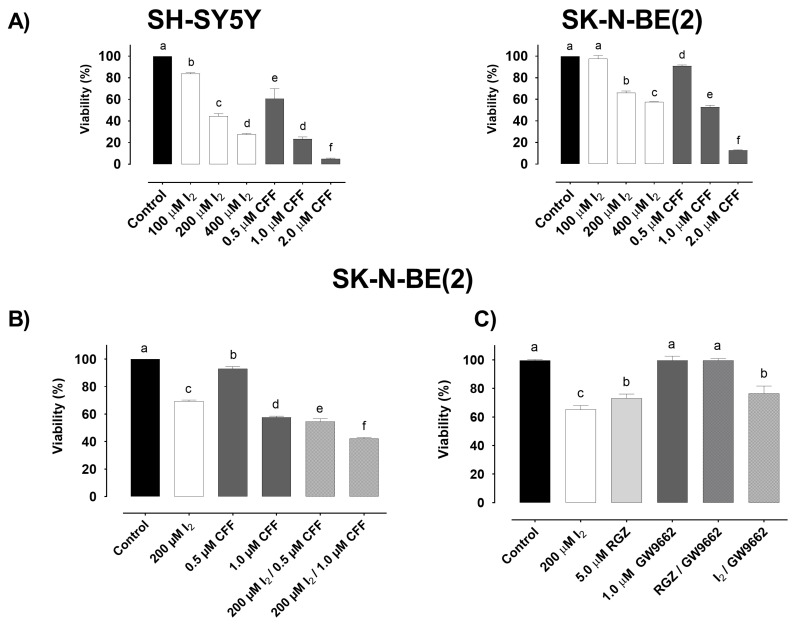
Effect of I_2_, CFF and their combination on the viability (exclusion dye trypan blue) of neuroblastoma cell lines. (**A**) Dose response of I_2_ and CFF in SH-SY5Y and SK-N-BE(2) cells. (**B**) Synergistic effect (CI < 1.0) of 200 µM I_2_ with 0.5 µM CFF (reduction 45%) and 1.0 µM CFF (reduction 25%) concentrations in SK-N-BE(2) cells. (**C**) Participation of PPARγ in the effect of I_2_ analyzed with rosiglitazone (RGZ; PPARγ agonist) and GW9662 (PPARγ antagonist) in SK-N-BE(2) cells. All experiments were carried out for 96 h. Data are representative of three independent experiments per triplicate and are expressed as the mean ± SD. Different letters denote statistical differences per group (*p* < 0.05).

**Figure 2 ijms-22-08936-f002:**
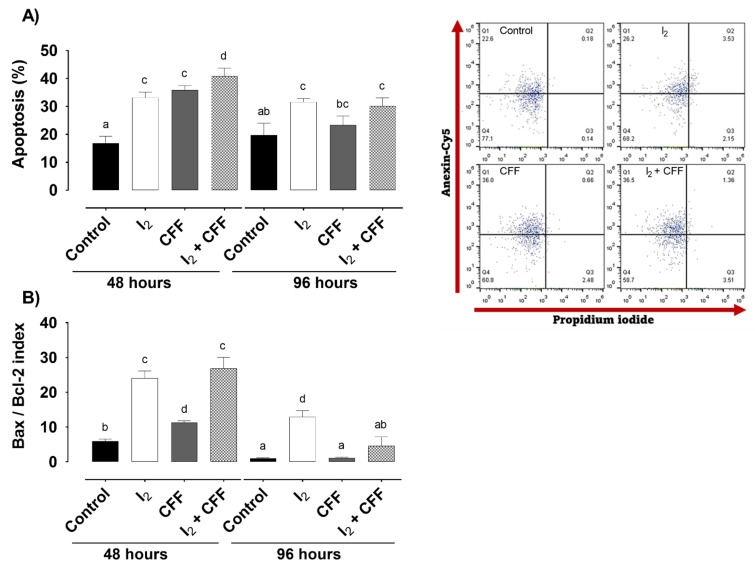
Apoptotic induction by I_2_ and CFF supplementation in the neuroblastoma SK-N-BE(2) cells. The cell line was supplemented with 200 µM I_2_, 1 µM CFF, and I_2_/CFF. (**A**) Apoptotic-positive cell percentage was evaluated with the Attune flow cytometer at 48 and 96 h; representative dot plots at 48 h and quantitative results are shown; (**B**) the apoptotic Bax/Bcl-2 index was assessed using real-time PCR at 48 and 96 h. Figures are representative of three independent experiments per triplicate. Data are expressed as the mean ± SD. Different letters denote statistical differences (*p* < 0.05).

**Figure 3 ijms-22-08936-f003:**
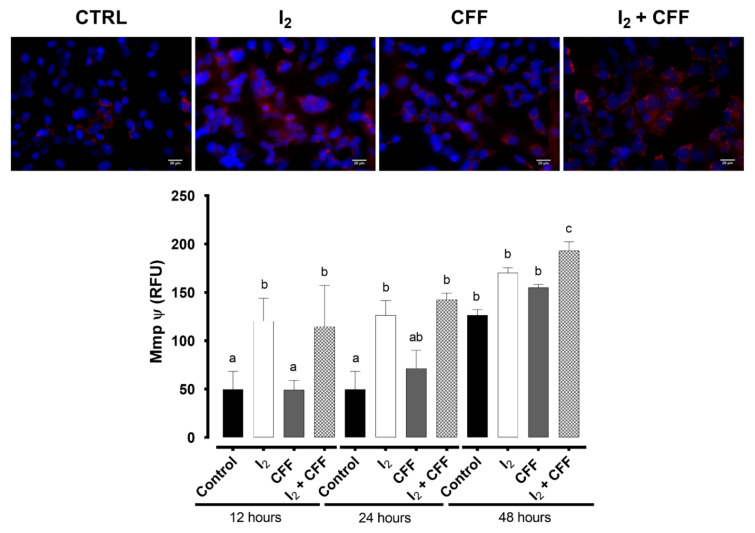
Effect of I_2_ and CFF on the mitochondrial functional state in SK-N-BE(2) cells. The cell line was supplemented with 200 µM I_2_, 1 µM CFF, and I_2_ + CFF, for 12, 24, and 48 h to evaluate the mitochondrial permeability (20 µM, scale bar). Representative micrographs (48 h) and bar graph for all times of the MitoTracker signal (Mmp ψ: mitochondria membrane potential change; RFU: Relative Fluorescence Units). Figures are representative of three independent experiments per triplicate. Data are expressed as the mean ± SD. Different letters denote statistical differences (*p* < 0.05).

**Figure 4 ijms-22-08936-f004:**
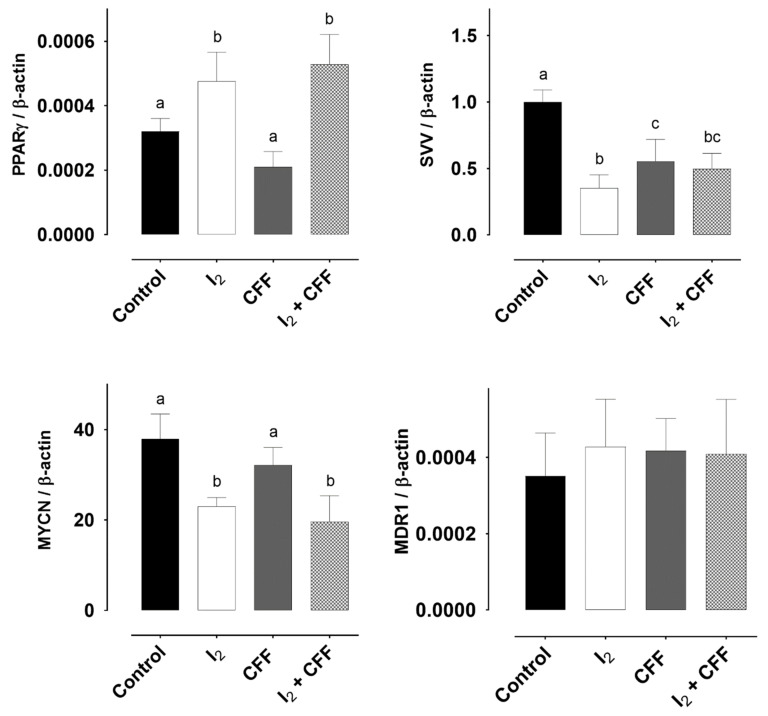
Effect of I_2_ and CFF on gene expression in the neuroblastoma SK-N-BE(2) cell line. I_2_ (200 µM), CFF (1 µM), and both (I_2_ + CFF) were supplemented for 96 h. Aggressive [neural MYC (MYCN)] and chemoresistant markers (Survivin (SVV), and multidrug resistance mutation 1 gene (MDR1)) were analyzed by RT-qPCR. PPARγ expression and protein content was analyzed by RT-qPCR (PPARγ/β-actin) and Western blot (PPARγ/Actin). Figures are representative of three independent experiments per triplicate. Data are expressed as the mean ± SD. Different letters denote statistical differences (*p* < 0.05).

**Figure 5 ijms-22-08936-f005:**
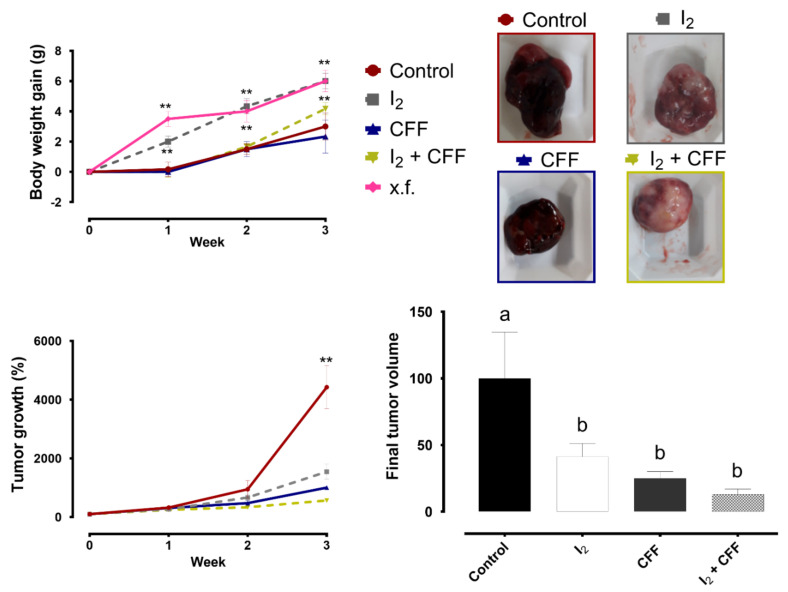
Effect of I_2_ and CFF on nude mice and xenografts. Nude mice with SK-N-BE(2) xenografts were supplemented with I_2_ (0.025%) and metronomic CFF (0.060%) in drinking water for three weeks. The line graphs show the body weight gain, % tumor growth and final tumor volume compared to the control group. The pictures are representative of the tumor bleeding appearance. Data are expressed as the mean ± SD (*n* = 4). Different letters and ** denote statistical differences for the control group (*p* < 0.01).

**Figure 6 ijms-22-08936-f006:**
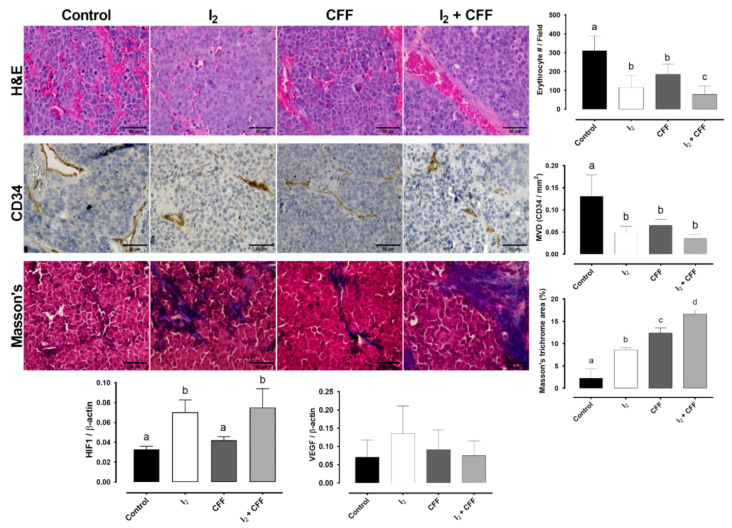
Effect of I_2_ and CFF on the histopathology of SK-N-BE(2) xenografts. H&E stain (40×). Immunohistochemistry of endothelial protein CD34 and quantification of mean vascular density (area CD34+/field). Masson’s trichrome stain and percent of positive fibrosis area. Epithelial (red) and collagen fibers (blue) (40×) (50 µM, bar graph). Hypoxia-inducible factor (HIF1) and vascular endothelial growth factor (VEGF) expression (RT-qPCR). Data are expressed as the mean ± SD (*n* = 4). Different letters denote statistical differences (*p* < 0.05).

**Figure 7 ijms-22-08936-f007:**
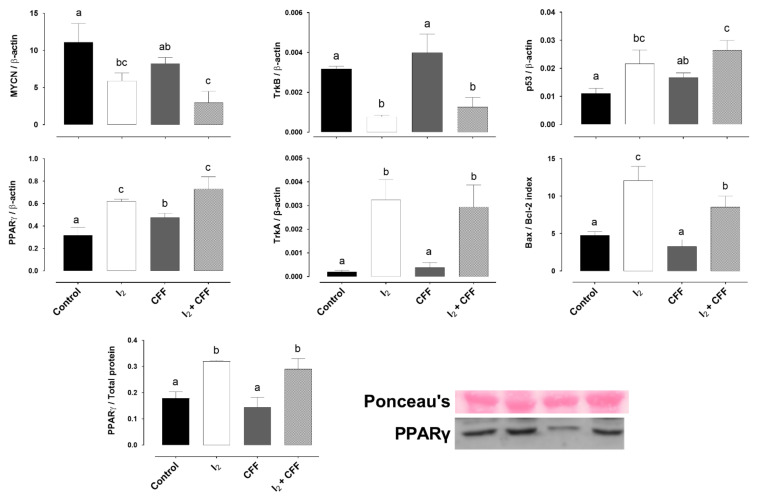
Effect of I_2_ and CFF on gene expression in neuroblastoma xenografts. Nude mice with SK-N-BE(2) xenografts were supplemented with I_2_ (0.025%) or the metronomic dose of CFF (0.060%) administered in drinking water for three weeks. Genes related to neuronal differentiation (PPARγ and TrkA), aggressiveness (MYCN and TrkB), and apoptosis induction (p53 and Bax/Bcl-2 index) were analyzed by RT-qPCR. PPARγ content was analyzed by Western blot. Data are expressed as the mean ± SD (*n* = 4). Different letters denote statistical differences (*p* < 0.05).

**Figure 8 ijms-22-08936-f008:**
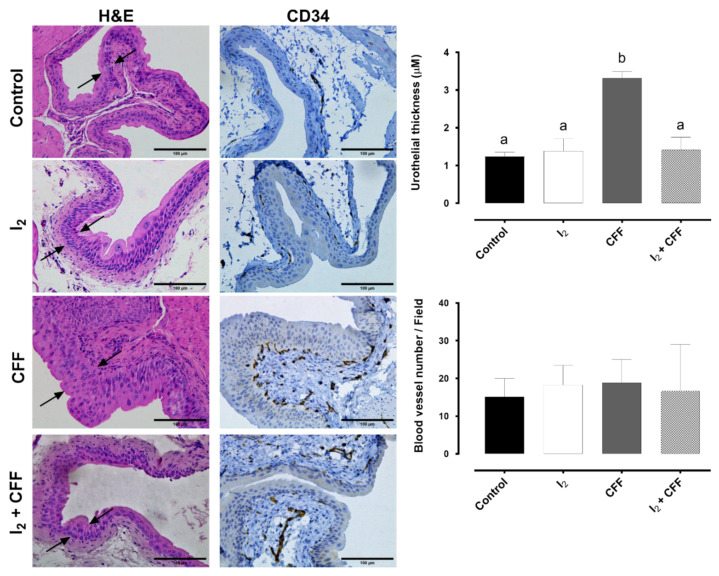
Effect of I_2_ and CFF on the bladder morphology of the chemoresistant neuroblastoma model. The figure shows the urothelial thickness indicated by arrows in H&E representative micrographs (100 µM, bar graph) and the vasculature quantification (blood vessel number/field) of CD34 immunohistochemical staining (representative micrographs and bar graph). Three sections per tumor sample were analyzed. Data are expressed as the mean ± SD (*n* = 4). Different letters denote statistical differences (*p* < 0.05).

**Table 1 ijms-22-08936-t001:** Oligonucleotide sequences.

Gen	Reference	Sense	Antisense	bp	Ta (°C)
TrkA	NM_002529.3	CATCGTGAAGAGTGGTCTCCG	GAGAGAGACTCCAGAGCGTTGAA	102	60
TrkB	NM_001007097.3	TCGTGGCATTTCCGAGATTGG	TCGTCAGTTTGTTTCGGGTAAA	231	60
PPARγ	NM_001354666.3	TCTCTCCGTAATGGAAGACC	GCATTATGAGACATCCCCAC	474	62
SVV	NM_001168.3	TTCTCAAGGACCACCGCATC	CCAAGTCTGGCTCGTTCTCA	126	60
MDR1	NM_001348945.2	GAGAGATCCTCACCAAGCGG	ATCATTGGCGAGCCTGGTAC	122	60
MYCN	NM_001293228.2	ACCCTGAGCGATTCAGATGAT	GTGGTGACAGCCTTGGTGTT	113	62
P53	NM_001126118.2	CCATGAGCGCTGCTCAGATA	GGGCACCACCACACTATGTC	124	60
Bax	NM_138764.5	AAGCTGAGCGAGTGTCTCAAGCGC	TCCCGCCACAAAGATGGTCACG	327	60
Bcl-2	NM_000633.3	CTCGTCGCTACCGTCGTGACTTCG	CAGATGCCGGTTCAGGTACTCAGTC	242	60
VEGF	NM_001025366.3	CTCGATTGGATGGCAGTAGCT	AGGAGGAGGGCAGAATCATCA	76	60
HIF1	NM_001530.4	TTGATGGGATATGAGCCAGA	TGTCCTGTGGTGACTTGTCC	128	60
β-actin	NM_001101.5	CCATCATGAAGTGTGACGTTG	ACAGAGTACTTGCGCTCAGGA	175	60

## Data Availability

The data presented in this study are available on request from the corresponding author.

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
