# Peer review of "Molecular Iodine/Cyclophosphamide Synergism on Chemoresistant Neuroblastoma Models"

_ijms, 2021, doi:10.3390/ijms22168936_

Round 1

Reviewer 1 Report

Dr. Aceves and colleagues describe a possible synergy between molecular iodine and cyclophosphamide in metronomic doses. Although some of the results are potentially interesting, there are weaknesses in the manuscript in its current form.

Major comment

1) The chemoresistance model is not clearly defined. The authors state that both neuroblastoma lines have a similar sensitivity to I2 (ratio 1.67), but for CFF at a similar ratio IC50 they state that this result confirms chemoresistance. In addition, a cell line less sensitive to I2 is also less sensitive to CFF. It is not entirely clear from the text whether it is chemoresistance in general or to CFF. Rather, I recommend referring to cell lines as cell lines with higher or lower sensitivity to CFF. In this context, it is also necessary to edit the title of the manuscript.

2) The authors, based on the comparison of cell viability after treatment with an inhibitor, resp. PARP activator and I2 claim that I2 acts primarily by influencing PARP. However, this experiment does not formally demonstrate that I2 affects PARP. It is necessary to perform a more specific experiment, at least to determine the expression of PARP gamma protein after the action of these substances, CFF a their combination. It is also appropriate to perform these experiments (viability and expression) with a line with lower sensitivity to CFF.

3) The authors found a lower percentage of apoptosis at longer incubation times with cytotoxic agents and a different trend when comparing apoptosis (annexin V) and bax / bcl-2, especially in the case of CFF. Is it possible to give an explanation?

4) I am not familiar with histochemistry, however, the description of the determination of the "vascular index" should be more accurate, resp. if CD34 is an endothelial marker, all vessels should be positive for CD34, ie the ratio will always be close to one.

Minor comment

CFF is a prodrug and its metabolism in vitro and in vivo is different. It is appropriate to discuss this in the context of the experiments performed.

Author Response

1) The chemoresistance model is not clearly defined. The authors state that both neuroblastoma lines have a similar sensitivity to I2 (ratio 1.67), but for CFF at a similar ratio IC50 they state that this result confirms chemoresistance. In addition, a cell line less sensitive to I2 is also less sensitive to CFF. It is not entirely clear from the text whether it is chemoresistance in general or to CFF. Rather, I recommend referring to cell lines as cell lines with higher or lower sensitivity to CFF. In this context, it is also necessary to edit the title of the manuscript.

The referee is correct. The response analyzes the two cells show equal sensitivity to both components I2 and CFF. We will correct this in the description of the results. It is also true that BE cells can be defined as lower sensitivity to CFF; however, in the literature, these cells are called chemoresistant because they are significantly less sensitive to many drugs (Coulter, et al, 2016); therefore, we believe that we can consider that our results of lower sensitivity corroborate its status as a chemoresistant cell line. We want to keep the title of our work that way.

Coulter DW, McGuire TR, Sharp JG, McIntyre EM, Dong Y, Wang X, Gray S, Alexander GR, Chatuverdi NK, Joshi SS, Chen X, Vennerstrom J. Treatment of a chemoresistant neuroblastoma cell line with the antimalarial ozonide OZ513. BMC Cancer (2016) 16:867

2) The authors, based on the comparison of cell viability after treatment with an inhibitor, resp. PARP activator and I2 claim that I2 acts primarily by influencing PARP. However, this experiment does not formally demonstrate that I2 affects PARP. It is necessary to perform a more specific experiment, at least to determine the expression of PARP gamma protein after the action of these substances, CFF their combination. It is also appropriate to perform these experiments (viability and expression) with a line with lower sensitivity to CFF.

We appreciate the referee's suggestion. We include the protein analysis (western blot) of the I2 and CFF response in tumors in the new version. A significant increase in PPARγ content was observed in samples from I2 treatments.

3) The authors found a lower percentage of apoptosis at longer incubation times with cytotoxic agents and a different trend when comparing apoptosis (annexin V) and bax / bcl-2, especially in the case of CFF. Is it possible to give an explanation?

It is normal behavior that in the long term (96 hours), the markers of early apoptosis (Bax / Bcl2) decrease because many of the apoptotic cells have already disappeared. The methodologies are complementary (flow cytometry and Bax / Bcl2), and although they are not strictly the same, the general behavior is similar: both I2 and CFF induce apoptosis.

4) I am not familiar with histochemistry, however, the description of the determination of the "vascular index" should be more accurate, resp. if CD34 is an endothelial marker, all vessels should be positive for CD34, ie the ratio will always be close to one.

The referee is right. To clarify this analysis, we redefined the quantification, and now we express it as the number of blood vessels per field.

Minor comment

CFF is a prodrug and its metabolism in vitro and in vivo is different. It is appropriate to discuss this in the context of the experiments performed.

We thank the referee for the observation. For the in vitro studies, the active metabolite 4-Hydroperoxycyclophosphamide was used, while the prodrug Cyclophosphamide was administered for the in vivo studies. To avoid confusion in the description of the results, both presentations are generically named CFF. Now we included this description in the material and methods section.

Reviewer 2 Report

Neuroblastoma is the most common extracranial neoplasm in children, with a high degree of malignancy, showing considerable phenotypic diversity and heterogeneous clinical behavior.  Molecular iodine (I2) induces differentiation and/or apoptosis in several neoplastic cells through activation of PPARγ nuclear receptors. The authors in a previous paper analyzed whether the coadministration of I2 and ATRA increases the efficacy of NB treatment. Theu proved, that I2 supplementation decreased the intratumoral vasculature. Authors suggest that the I2 + ATRA combination should be studied in preclinical and clinical trials to evaluate its potential adjuvant effect in addition to conventional treatments. In this study, authors analyzed the effects of I2 and CFF on the viability (culture) and differentiation of Nb cells. The results showed that it is a dose-dependent antiproliferative effect, with I2 increasing the sensitivity of Nb cells to CFF and  significantly inhibiting Nb xenograft growth, and reducing the risk of proliferation. The authors have smartly presented the reducing of proliferation (Survivin) and activating of apoptosis in NB. They found that I2 decreased the expression of markers of malignancy (MYCN, TrkB), gave blood vessel remodeling and increased differentiation signaling (PPARγ and TrkA).

Recent evidence has demonstrated high expression of somatostatin receptors in neuroblastoma (NB) cells. Combination of 177Lu-DOTATATE with chemotherapeutic agents might achieve worthwhile responses with low toxicity, encouraging survival in NB patients who have relapsed or are refractory to conventional therapy, including 131I-MIBG therapy. (Clin Nucl Med. 2021 Jul 1;46(7):540-548.)

Authors are asked to make a number of clarifications and corrections before deciding to publish:

  1. please write clearly what was the control in the experiments conducted
  2. in gene expression studies, please complete the results - primary studies - before culture - results of gene expression studies and results of gene expression after culture without I2 and CFF
  3. in further stages of experiments, e.g. concerning mitichondrial membrane, immunohistochemistry, please add the results of preliminary tests before culture and administration of I2 and CFF, as well as the results after culture without I2 and CFF
  4. the statistical analysis is not clear at all - please show in the additional materials a table with individual test results, with mean and deviation results, median etc. for all stages of the experiment - preliminary results before culture and after administration of I2 and CFF, and results after culture without I2 and CFF
  5. tests - cultures are conducted in different concentrations of CFF - results and figures do not show it - please supplement
  6. please state explicitly whether analyzed and compared results were significant and present test value and p
  7. After completing the results, please review the discussion

The paper is interesting and raises new issues regarding the novel use of drugs in refractory and relapsed neuroblastoma, but needs significant revision.

Author Response

Recent evidence has demonstrated high expression of somatostatin receptors in neuroblastoma (NB) cells. Combination of 177Lu-DOTATATE with chemotherapeutic agents might achieve worthwhile responses with low toxicity, encouraging survival in NB patients who have relapsed or are refractory to conventional therapy, including 131I-MIBG therapy. (Clin Nucl Med. 2021 Jul 1;46(7):540-548.)

We thank the referee for this reference. It is an interesting study, and we included it in the introduction

Authors are asked to make a number of clarifications and corrections before deciding to publish:

  1. please write clearly what was the control in the experiments conducted

The control group is conformed by the cells treated with similar culture conditions plus 50 µl of distilled water (vehicle of I2 and CFF) for the same times. The graphs indicate the percentage of change of the treatments versus the control group at 96 hours. Now we include as supplementary material the data obtained during the hours of culture (24, 48, 72, and 96 hours) for both compounds I2 and CFF. Also we include a phrase indicating “control groups were followed at the same times using deionized wather as treatment (vehicle, I2 and CFF).

  1. in gene expression studies, please complete the results - primary studies - before culture -results of gene expression studies and results of gene expression after culture without I2 and CFF

The control groups corresponded to cells seeded simultaneously and kept for 24, 48, 72, or 96 hours without treatment. The gene analysis was performed in all cells at 96 hours. There are no cells before and after treatment

  1. in further stages of experiments, e.g. concerning mitochondrial membrane, immunohistochemistry, please add the results of preliminary tests before culture and administration of I2 and CFF, as well as the results after culture without I2 and CFF

The experiments in this section were done in the same manner as previously described; the controls are carried out in the same hours as the treatments. There is no before and after treatments.

  1. the statistical analysis is not clear at all - please show in the additional materials a table with individual test results, with mean and deviation results, median etc. for all stages of the experiment - preliminary results before culture and after administration of I2 and CFF, and results after culture without I2 and CFF.

The complementary material shows the monitoring of cell cultures for 24-96 hours for each group. The statistical tests used are mentioned in the statistical analysis section as well.

  1. tests - cultures are conducted in different concentrations of CFF - results and figures do not show it - please supplement

Figure 1 and the supplementary figures show the different concentrations of I2 and CFF to calculate the IC50, subsequent experiments were done at concentrations close to the IC50 of each (200 uM and 1.0 uM, respectively).

  1. please state explicitly whether analyzed and compared results were significant and present test value and p

In the material and methods section, there is a section with this information.

 4.5. Statistical analysis

Data for in vitro experiments are the media of three independent tests in triplicate. In vivo, four animals per group were used. Tissue analysis for PCR and western blot is the average of 4 samples, and three sections of each tumor were used for immunohistochemistry. Statistical analysis was performed by one-way ANOVA followed by Tukey’s test for analysis between groups. The progression of xenografts was analyzed One-way ANOVA as a post hoc test. Values with P<0.05 were considered statistically significant.

  1. After completing the results, please review the discussion

The manuscript has been revised, and several details have been included.

The paper is interesting and raises new issues regarding the novel use of drugs in refractory and relapsed neuroblastoma, but needs significant revision.

Thank you very much for the useful suggestions and deep review

Round 2

Reviewer 2 Report

The experiments were planned without including the analysis of results before and after culture and administration of I2 and CFF. It is a pity that there is no reference point, to some extent this weakens the significance of the results obtained. Nevertheless, thank you for the clarifications and corrections made in the text.